# Ideal Mechanization: Exploring the Machine Metaphor through Theory and Performance

**Amy LaViers** 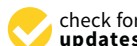

Mechanical Science and Engineering Department, University of Illinois at Urbana-Champaign, Champaign, IL 61801, USA; alaviers@illinois.edu

**Abstract:** Models of machines, including the increasingly miniaturized, digitally controlled machines of modern computers, inform models of human and animal behavior. What are the impacts of this exchange? This paper builds on theoretical discussion to produce an artistic exploration around this idea. The paper uses known limits on computation, previously proved by Turing, to model the process of mechanization, machines interacting with an environment. This idea was used to inform a live performance that leveraged a theatrical setting emulating an ideal mechanization machine, audience participation with their bodies as well as their personal cell phones, and readings of academic papers, which is also presented. The results of this work is a shared exploration of when human experience fits machine-based metaphors and, when it does not, highlighting distinct strengths and questioning how to measure the capacities of natural and artificial behavior.

**Keywords:** interactive performance; robotics; human motion; Turing Machines

## 1. Introduction

Computers and robots are increasingly attributed with behaviors of natural systems. The fields of machine learning and artificial intelligence often lose their synthetic modifiers: "machine learning" becomes "learning", "robotic arm" becomes "arm", and so on. From some points of view, such comparisons may be appropriate: if a computer beats a human player in chess, we may say that the computer played the game of chess successfully. However, for those outside of engineering disciplines or without knowledge, of the details of the system, it may be easy to then conclude that the computer processing used was *the same or even superior to* the human player. Moreover, in the broader scientific community, technical models become representation for analogy to biological function.

For example, the bio-inspired structure of the networks in AlphaGo that beat human Go players suggests a direct comparison of machine similarity and superiority to humans (Silver et al. 2016) and has been discussed in popular media as just exactly that (Cheng 2016; Metz 2016) without the same modifiers and subtlety used in the technical report (LaViers 2017). However, as technical readers understand, the system cannot imitate *most* features of human activity. Moreover, in controlled factory settings, robots can often outperform humans, offering greater precision, higher-payloads, and consistency in repeatable tasks (Kinova Robotics 2017; Rethink Robotics 2018; Universal Robots 2016). That is, from the point of view of Newton and mechanics, robots are superior mechanical devices—yet they cannot replicate many features of human movement. What quantitative models can explain this? One answer may come from Turing (1936) whose work has shown that machines have a limited, though infinite, set of behaviors available to them.

Increasingly, advancements in technology are used as explanatory models for biological function (Turner 2013). Neural networks have been proposed as models for brain activity (Churchland et al. 2016). Minimal coding schemes which drive data compression have also been used to model neuron communication (Spratling 2017). Logic gates have been used to describe

cell function (Zah et al. 2016). Distributed control algorithms inspired from nature also have been used as explanatory models for flocking behavior (Nabet et al. 2009). Optimality has been proposed as a model for motion generation with respect to minimizing jerk along a movement trajectory (Flash and Hogan 1985) and where optimal control may provide a "unified theoretical framework" for sensorimotor systems (Todorov and Jordan 2002). These examples illustrate how our understanding of machines and mathematics forms a basis for understanding the natural world.

Imitating the movement of biological organisms, a subfield of biomimicry (Bar-Cohen 2005), using similar, continuous metrics for success, has been a topic in animation (Reynolds 1999) and robotics (Egerstedt et al. 2005; Powell et al. 2012). However despite an explosion of computing power, including cloud-based devices, robots do not thrive in dynamic environments nor can they recreate the social behaviors of humans, even under teleoperation (Yanco et al. 2015). Even for relatively simple organisms, such as fish, developing robotic counterparts has proven challenging (Marras and Porfiri 2012) with initial success in creating robotic counterparts that school with real fish (Landgraf et al. 2016; Swain et al. 2012). While these robotics fish are some of most successful robotic confederates, even simple machines such as Braitenberg's (1986) vehicles have been shown to be *expressive* to human viewers, indicating that perfect imitation is not needed for meaning-making. One of the best studied organisms, *C. elegans*, have had their 302-neuron network mapped out; moreover, a linear behavioral model describes much of the behavior they exhibit in laboratory agar plates (Stephens et al. 2010). In human tasks, movement quantification is difficult and limited to specific tasks, e.g., drawing (Del Vecchio et al. 2003).

The work of Turing (1936) and Elgin (2010) are at direct odds in their treatment of human behavior. Elgin (2010) posited that "If non-propositional items can advance understanding, then the thesis that dance advances understanding has some chance of being correct". Computers, however, rely on propositional structure to complete computation, and Turing's, and subsequent, comparisons of these models to human experience have been challenged by theorists like Godel (Gödel 1972; Shagrir 2006). Shannon (1938) showed how Boolean Logic can be implemented in circuits; similarly, modern robotics research frequently leverages Linear Temporal Logic (LTL) to produce robot behavior (Belta et al. 2007). Likewise, the structure of Turing's automatic machines is limited to systems with distinct finite states governing system behavior. This is one abstraction of many that helps us understand what computation is and what it is not. Understanding the notion of *subjective experience* is ongoing work by biologists and philosophers that is influenced by the structure and terminology of control algorithms for robots (Godfrey-Smith 2016).

Elgin (2010) wrote that dance advances understanding through variable motion profiles that exemplify ideas, both literal and abstract; moreover, the innovation of choreographers is to create new behaviors. Therefore, dance theory is a place to look for understanding about how information is carried in the motion of a moving body to a human viewer. Moreover, dance seems to imply that variation and complexity of motion profile may be important components of the biological function of movement. Indeed, many species besides humans are known to cue through complex "dances" (Soma and Iwama 2017). Thus, a way to count the number of behaviors accessible to natural systems may be an important measure of their performance, which is distinct from the mechanical measures previously listed, where the performance of robots has exceeded human abilities in the range of torque and velocity and in the precision and repeatability of movement.

A comparison by Changizi (2003) of observational studies of the complexity of behaviors exhibited by an organism and number of muscle types discovered in that organisim shows a positive correlation between encephalization quotient and number of behaviors exhibited. In particular, this work plots the number of exhibited behaviors, $E$, against the number of muscle types, $C$, and finds a power law relationship between the two, i.e., $E \approx C^3$. Changizi compared this to English language where words and exhibited sentences have a power law relationship with a factor of 5; he posited that the power law factor of 3 found in his study may be higher for more complex animals, such as humans. This work shows the non-Turing behavior of animals and is related to the quantified relationships

that other types of studies have been able to create, e.g., relationship between size and speed of organism (McMahon and Bonner 1983). This work suggests an evolutionary advantage to behavioral complexity—or, that expressive behavior serves a functional purpose.

The Laban/Bartenieff Movement System outlines a duality between function and expression in motion (Studd and Cox 2013). For example, consider an agent moving angrily through a living room, thrashing its arms wildly and taking heavy, sure-footed steps. On the other hand, an agent moving through a jungle may need to exhibit the same motion, but it would not be perceived as angry in this new context. The concept of *expression* is foregrounded in examples such as the agent moving angrily through a living room. Its dual, *function*, is foregrounded for the same motion profile in the jungle. However, in both examples, the opposite ideal is still occurring: in the living room, the agent is succeeding at its task of informing a counterpart; in the jungle, the agent is simultaneously communicating capability in the jungle setting to a human viewer. Thus, functional models of motion generation, where minimum energy, or maximum speed, produce motion behaviors, break down when context and communication are considered. This is analogous to function and form as abstract ideas that reinforce one another in product design where product designers have also posited the idea that physical objects communicate with human users (Crilly et al. 2004) and points to the idea that roboticists need to similarly begin factoring in the human experience of systems when designing their movement (LaViers 2019).

Artistic exploration with robotic systems (Cuan et al. 2018; LaViers et al. 2014, 2018), including cell phones (Toenjes et al. 2016) and Turing's work (Gow et al. 2014), in theatrical settings, alongside professional dancers, inform the point-of-view presented in this paper. When dancing with robots, dancers report a lack of variability in the motion of these systems; correspondingly, choreographers have found the platforms to be frustratingly limited in their expressive capacity. The *functionally efficient* approach used by roboticists to design robot movement may be part of this experience. In contrast, dancers care about *expressively rich* movement, working to create many options for movement behaviors in their bodies through extensive training. Across many classes they work to increase their range of motion, develop ability to coordinate multiple actions simultaneously and in sequence, and hone their execution of different textural qualities. By broadening and maintaining their movement options, they become more versatile instruments. Is there a limit on the options they can pursue? What is it?

In robotic systems, changes to hardware and/or software can expand the capacity of a given system to exemplify ideas. For example, efforts to bridge human notation of movement to robotic motion specification are covered in (Laumond and Abe 2016), which covers algorithmic development for humanoids based on Labanotation (Salaris et al. 2016) and where Benesh notation experts explicate the challenge, of recording and translating human motion (Mirzabekiantz 2016) and dancers lament the nuance lost in motion notation, suggesting "impossibility" (Challet-Haas 2016). On the hardware side, while a modular snake robot, e.g., (Wright et al. 2007), may not be anthropormorphic, its variable gaits provide the ability to indicate a change in internal state, which is vital for expression and communication in a theatrical context. However, known limits on machine behavior (Turing 1936) suggests we can imagine behaviors not implementable on machines. Are these "imagined" behaviors possible in natural systems? Can human bodies express more ideas than machine bodies or do limits, such as the one proved by Turing, hold?

Section 2 outlines an abstraction for mechanical motion by leveraging the Turing Machine formulation of computation. Section 3 provides a development of artistic themes and compositional questions inspired from this formulation that are put into action in a live dance performance described in Section 4. The goal of this performance is to provide embodied experiences to audience members, giving them access to an academic debate in a short amount of time. Concluding remarks and future directions for this work are given in Section 5. The contribution of this paper is not in theoretical computer science or the philosophy of dance; these are two fields from which this work is building and applying existing ideas. The contribution of this paper is presenting work from these fields—as well

as biomimetics and robotics—alongside one another in a performative context. The paper uses this existing academic discourse to motivate the creation of a particular performance. This performance is only one of a myriad of creations that could happen in response to the body of literature presented. It is the aim of the paper to encourage other artistic exploration in this vein, and the paper is written for an interdisciplinary audience in order to facilitate this possibility.

## 2. Mechanization: An Ideal, Discrete Process with Limits

Discussion in the previous section reviews the literature on the movement of human and machine bodies. This section introduces an abstraction for systematized movement that is the inspiration for the performance described in the following two sections. The discussion in this section more explicitly outlines how movement is inherently part of computation, using a model of computation to outline an ideal process for mechanization. This discussion uses an established abstract model of computation for this exploration (a Turing Machine) and does not exhaustively review every aspect of this large field. However, it is important to note that Turing Machines are not real computers, and, thus, the model for mechanization proposed here does not describe real robots. Instead, Turing Machines are one abstract model of computation that here provides one way to *count the number of behaviors possible in a class of artificial systems.*

In (Turing 1936), Turing outlined an *a-machine*, a machine with a finitely complex mechanical head along an infinite tape where symbols can be stored. The abstract machine requires the current configuration of the head, a list of basic instructions that tell the machine what to do in that configuration, and the complete configuration (state) of the entire system.

The components of an a-machine are given by the following list, loosely following (Immerman 2016):

- a finite set of $n$ machine states $Q = \{q_1, ..., q_n\}$;
- a finite set of $m$ symbols $\Sigma = \{\sigma_1, ..., \sigma_m\}$, e.g., $\Sigma = \{0, 1, \epsilon\}$, where the result of machine computation, a computable number, is recorded in binary with a blank option, $\epsilon$;
- an infinite "tape" where these symbols are recorded, comprised of cells $c_1, c_2, c_3, ...$, which is often pre-populated with a finite sequence of symbols that generate programmed behavior when the machine is in operation;
- current position along the tape, cell $c_h$, where $h \geq 1$; and
- a transition function $\delta : Q \times \Sigma \mapsto Q \times \Sigma \times \{-1, 0, 1\}$, which determines at a given state $q_i$ for a given scanned symbol $\sigma_i$ in $c_h$ how to update the position of the head $h$, i.e., it moves left, stays in place, or moves right.

Innovations such as stored program architecture and clocking have been important to developing real, modern computers but do not change a central premise of Turing's paper. Specifically, he defines the class of numbers that can be computed by a properly formed (circle-free) machine to be enumerable (infinite but countable). That is, there are numbers that we might imagine that cannot be computed (e.g., irrational numbers without algebraic formulas for computation). This is seen through application of Cantor's diagonal process, which shows that the correspondence between natural numbers and computable numbers is one-to-one (or that the set of real numbers and computable numbers is not one-to-one) due to an inescapable recursive loop that traps an a-machine checking its own description number (this is known as the Halting Problem) (Petzold 2008). That is, we can imagine numbers that a-machines cannot compute, which means we can imagine behaviors that a-machines cannot perform. Other theoretical formulations of computation, such as Wegner's (1997) work with the abstraction of interactions, which does not change the set of computable numbers (Prasse and Rittgen 1998), are not considered in the creation of *A Machine*, where the exploration is centered around the cardinality of distinct behaviors that are enumerated by this set. New architectures that improve the swath of computable numbers, such as Siegelmann's (1995, 2013) Super-Turing formulation, and her extensions in modeling natural systems with analog devices, highlight how discrete computing devices, such as

modern cell phones, where Turing's definition of a computable set of numbers is a fundamental limit, cannot capture the behavior of many natural, chaotic systems.

To establish a way of thinking about machine movement (mechanization), invert the a-machine, establishing an æ-machine. In this abstraction, the idea of a physical workspace replaces Turing's idea of "scratch paper" where computations could be worked out. An ideal mechanization machine will be able to complete tasks in the physical environment using extra workspace as needed. Here, the motion of the Turing Machine is foregrounded: its motion, which occurs in discrete units that may be called *motion primitives*, the behavior of interest. We are now focusing on the motion, right to left, of this abstraction, and we want to know: Can this device produce any arbitrary pattern of right to left movement? This is a discretized model that may be analogous to how a human or robotic artisan will use a workshop table during their work, placing part of a product off to the side while working on another element, using this tool or that to complete various steps, and increasing the size of their workshop as needed. Similarly, the tape need not be one dimensional; this machine can perform actions inside its workspace, layering simple actions in sequence to produce desired effect on the environment. Thus, we can define the components of an æ-machine as follows:

- a finite set of $n'$ states $Q' = \{q'_1, ..., q'_n\}$;
- a finite set of $m'$ actions, or *motion primitives* $\Sigma' = \{\sigma'_1, ..., \sigma'_m\}$, e.g., $\Sigma' = \{flexion, extension, \epsilon'\}$, where the result of machine mechanization is executed as either moving, moving in the opposite direction, or doing nothing, $\epsilon'$;
- an infinite "workspace" where these actions are executed, comprised of cells $c'_1, c'_2, c'_3, ...,$ which may be pre-populated with a finite set of primitives (or tools) that generate programmed behavior when the machine is in operation;
- current position in the workspace, cell $c'_{h'}$, where $h' \geq 1$; and
- a transition function $\delta' : Q' \times \Sigma' \mapsto Q' \times \Sigma' \times \{-1, 0, 1\}$, which determines at a given state $q'_i$ for a given motion primitive $\sigma'_i$ in $c'_{h'}$ how to update the position in the workspace $h$, which might be envisioned as a one-, two-, or three-dimensional "tape".

What was a computation process (a sequence of logical symbols manipulated in an abstract, memory-like space) is now a mechanization process: a sequence of motion primitives executed in a discretized environment. This sequence can likewise be represented as a number—one from Turing's set of computable numbers—showing the infinite, but enumerable, action sequences possible to be executed by æ-machines. Thus, æ-machines (an idealization of robots) have the same fundamentally limited capacity as a-machines (an idealization of computers). That is, *they cannot produce all the behaviors we might arbitrarily design*. This means there are sequences of machine motion that we might imagine that are not mechanizable—those which correspond to a number in the set of uncomputable numbers, which does not include the set of real numbers. Similar to how Turing established subroutines to build his Universal Machine, we can create more complex behaviors of motion primitives that occur in sequence, acting as a *tool* in the workspace. In practice, that tool could be "software" (a stereotyped, preprogrammed gesture or action) or "hardware" (an end effector attachment as a CNC machine selects distinct cutting tools).

## 3. Translating Machine-Based Metaphor to Elements of Live Performance

Thus, an abstract theory in which the behavior of both computers and robots is described by the set of computable numbers has been presented. This ideal holds regardless of the specific structure of an implemented system. In computers, more transistors do not change the fundamental capacity of behavior (the set of computable numbers), and, likewise, changing the mechanical complexity of a robot may not change the fundamental capacity of behavior. The Turing abstraction is not a literal definition of how to build a computer, and it is not meant as such here. Therefore, the theatrical goal is not to set up a literal computer or a literal robot but for audience members to feel the texture of these

ideal models. In doing so, the piece will explore the very real ways that humans out-perform modern day machines and vice versa.

This develops questions that are explored through artistic practice and performance: Is the machine metaphor apt for human performance? Are humans limited by the same idealized abstraction as the one Turing outlines for machines? Does non-propositional logic guide human understanding and experience? This paper does not provide a quantitative answer to these questions, but instead poses them as motivation and presents an exploration of them through art and embodied experience. Modern machines, which are in one specific, definite state at a time and take advantage of computational resources that are not spatially co-located have a form that is very unlike our own. Thus, the piece will include themes of calling from physical spaces that are not co-located with the performance, accumulating complexity and growth in our representations, and in binary logic, which underpins modern machines.

### 3.1. Intended Audience

It is anticipated that each audience member would bring with them a powerful, mobile discrete computing device in the form of their personal cell phone. Increasingly, we have antagonistic relationships with these devices (Jenaro et al. 2007); thus, a goal for this audience is to reframe and deepen their associations with these tools, which can be a theatrical proxy for thinking about computation more broadly, but is also a point of access to personal feelings of attachment, anxiety, and fear (Howe 2017; Mokyr et al. 2015) that many modern audience members bring with them to the start of the show.

Moreover, popular descriptions of machines frequently emphasize their human-like capabilities (e.g., Baldwin 2019; Berboucha 2018; Madrigal 2018; Mae et al. 2018). These descriptions contribute to real anxiety about the future for the public many of whom have little experience programming, an experience which affords a different perspective on how machines work to technically trained members of society. It has been estimated that in 2018 there were 23 million programmers worldwide (Garvin 2018), while 4.57 billion people were predicted to own a cell phone that same year (eMarketer 2015). Thus, the performance is motivated by the fact that as many as over half of the people in the world carry a type of re-programmable machine that they have no experience in programming. This piece is accessible in the sense that audience members do not need to pass a math test or compile a computer script to participate.

### 3.2. Rational for Selected Elements

Development of a dance performance began with the image of a simple system moving through an idealized, discretized workspace in which new tools, developed through compounded hardware units and nested software routines, are collected. More simply, the image can be thought of as a robot walking out into the world picking up tools, ever-increasing its own instantaneous complexity through new additions to the system. Likewise, humans are constantly developing new tools to increase our abilities. Often philosophy is locked in hard to access academic papers (specifically of interest in here is the philosophy of meaning in dance and how it may be generated through non-propositional structures (Elgin 2010) versus philosophy of decidability (Turing 1936), which relies on propositional structures), and one goal of the performance was to expose these ideas to a broad audience for examination. Three main themes were distilled as guidelines for creation of a piece entitled *A Machine*, a pun on Turing's term *a-machine*. The goal with these themes is to evoke the dark, rigidly structured internal world of computers, made visible and experiential through embodiment, transporting audiences to the quirky, unnatural innards of machines.

(1) *Workspace/Workshop Setting.* The setting of the piece would be both abstract and concrete: a representation of an idealized, discrete workspace combined with an artisan's workshop. Tape segmenting the area of a dance floor, forming a substrate onto which continuous movements may be seen as discrete *motion primitives* and an apron as a costume, referencing an article common to many

craftspeople, are two ways the performance, implemented this idea. The physical limits of modern machines and humans would be contrasted by bringing extant, everyday machines into the setting in a formal way, bridging the intimate relationship humans have with their machines, e.g., personal cell phones which were expected to be on the person of audience members attending the show. Movement composition explored the idea of finite, bodily limits and pushed to show the multiplicity of human bodily forms. The use of repetition and compositional structures that implied invisible rules were used, material which reflects an *embodied experience* of working with machines, which often leverage repetition in automated tasks and require formal instructions that *feel*, in practice, unnatural.

(2) *Rigidity of Logic-Based Statements.* The piece would explore how forced true/false statements can break down in characterizing human thinking and experience. Using statements that are hard to answer with a single bit of information, e.g., "Yes" or "No", highlights this limit of computers: while increased complexity can represent levels of gradation and many categories, at their heart "kind of" is not native to modern-day machine function. On the other hand, describing the human experience involves the notion of *subjectivity* where humans frequently exhibit indifference and multiplicity of thinking. This echoes Godel's ideas about modeling human thought with finite, discrete states. One way that this distinction can be *felt* is in the process of programming a computer, but this is a rarefied experience. As discussed in the previous section, most people alive in the world today have not written a computer program; thus, the performance would leverage theatrical techniques to provide an accessible way to *experience something such as the distinction between imparting behavioral instructions to a machine versus to a human.* Asking audience members to answer ridiculous statements with "true" or "false" similarly forces the experience of fitting into an uncomfortably finite structure, which may not be native to biology.

(3) *Segmented versus Continuous Experience.* Finally, the piece would explore rhythms and concepts native to machines versus humans. Using electronic noises throughout the performance is one way this texture was created in the soundscape. Readings from Turing's (1936) work highlight how series of numbers and letters, which are meaningful to computers and in mathematical proof, sound manic when spoken aloud and moved. The cellular segmentation of the space would be used to create a breakdown in natural gait and create contained movement phrases. Successfully stepping over tape is a discrete phenomenon, while the experience of moving a foot is continuous. This is reminiscent of a possibly familiar activity of walking without stepping on cracks in the sidewalk, but also can serve as an embodied version of the mind-bending process of creating discrete criteria for sorting numbers into arrays (for example).

## 4. Performance of *A Machine*

The resulting piece, *A Machine*, was performed [details redacted for review]. The live performance allowed for a physical exploration of the theoretical ideas discussed here, a chance for reflection and feedback from live human bodies, and an outreach and sharing activity. The performance was not a formal research study, but future instantiations could use this initial showing to construct response collection under differing conditions, such as demographic information. Each section of the performance is discussed in this section. Figures 1–6 present images from the performance of *A Machine* on [date redacted] throughout this section complement this discussion.

### 4.1. Show Advertisement

The show was advertised with the following description, which was designed to make audience members aware of the nature of the show, and possibly arrive with heightened awareness and anticipation about the type of participation that would be required of them.

Can machines exhibit the same behaviors as humans? In what ways is your phone superior to you? How can you outperform a robot? Can you live without your tools? Are you a machine? This showing is part of an in-progress academic paper that requires embodied

inquiry. Come be a part of research on 19 December 2018 in Dance Rehearsal Krannert (DRK) in the Krannert Center for the Performing Arts at 5 p.m. Seating is limited and participation is required.

### 4.2. Entry and Pre-Show: Personal Machine Collection and Seating

The performance space was demarked with long strips of thick grey gaffers tape, arranged in an irregular grid. The grid points were densely packed near one end and sparse at the other. Upon 13 intersections of tape, 13 square, numbered cushions of green foam were placed. The numbering system used was base 16, or hexademical (such that 10–13 were written as "A", "B", "C", and "D"). This set up is shown in Figure 1. This choice references the numbering systems often used in assembly languages where commands correspond to directly to transistor-based hardware; moreover, the unexpected choice served to cause audience members to try and "figure out the system" even before the piece began.

Upon arriving to the performance, audience members were presented with two seating options. Either they could sit in a section of chairs behind the performance space or on the floor on numbered cushions inside the space for a participatory experience. Participants were offered these seats on a first-come, first-served basis. Several audience members did not want to sign the photo release required for this or surrender their personal property for the duration of the show. These audience members (not pictured in this paper) were seated in rows of chairs outside of the performance area.

Those who elected to participate surrendered their personal cell phones, providing their phone numbers and turning the ringers on their phone *on* and *up* as well, and filled out a photo and video release. Each of these audience members (13 in total; a number that was based on the setup of the room) were assigned a number and headed out into the space to their assigned position, corresponding to a numbered cushion (shown in Figure 1). Meanwhile, their cellphones were wrapped in correspondingly numbered foam, secured with two rubber bands, and placed in a basket. The program note (see Appendix A) was displayed on a projector for audience members to read as they entered the space (shown in Figure 1). The bright light of this mostly cool-white projection contrasted the shadowy, warmly lit performance space. Music, entitled "Black Energy" and "In the Beginning", from the album *Planetarium* composed by Sufjan Stevens, Bryce Dessner, Nico Muhly, and James Mcallister played during this time.

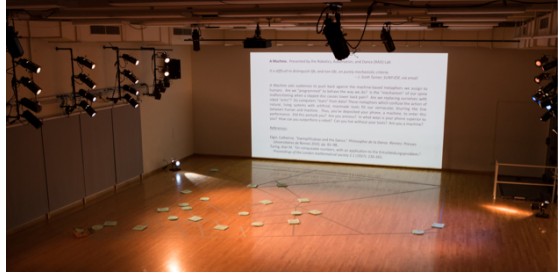 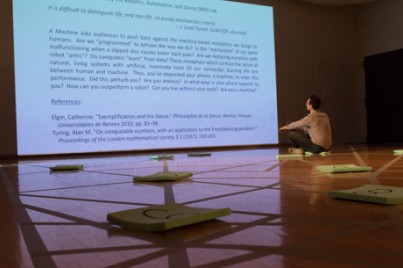 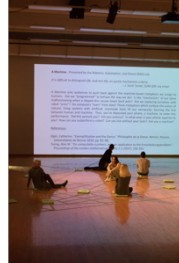

**Figure 1.** Seating audience members before the show. Low lighting and a sparse "environment" for participating audience members to sit in creates a stage that is a studio-like setting that feels lifted from the pages of a geometry textbook: natural wood grain contrasts thick industrial grey tape laid in a linear mesh; warm theatrical lights contrast cool white from a projector; and the voices of arriving audience members contrast electronic ambient noises. Photos by Natalie Fiol.

### 4.3. Part 1: Machine Metaphor and Workspace Setup

Once all audience members were seated, the performance began with a video of the performer writing numbers 1–13 in hexadecimal while reading excerpts of Turing's paper, outlining the properties of his "a-machine", now known as a Turing Machine (Turing 1936). These excerpts highlighted the finite properties of machines, Turing's proof that leverages a human computer writing symbols on a linear sheet of paper, and several moments where the reading of his Description Numbers created long strings of numerals and letters that provided the live performer a strange, monotone rhythm with

which to move. The motion of writing the numbers was regular, informed by the predictable shape of each symbol; however, it also grated against the descriptions of the automatic machines described by Turing as the performer made irregular choices in filling in some sections of the symbols more than once, painting with a black marker on rough foam.

After the video played for several moments, the performer entered the space, stepping only into the open spaces of the grid, carrying each individually wrapped phone, and placing it in a location on the grid. The performer wore an apron and rubber bands on both wrists, placing her in a workshop environment where the rubber bands were a tool in her workbench. As each phone was unwrapped, she reinforced this image by placing the removed rubber bands onto the cadre of existing bands on each wrist. This activity has a natural, discrete propositional notion associated with it, e.g., the rubber band is off or on the wrist, and the ritual served to diversify the motion tasks, or, loosely, *primitives*, seen in this section of the piece.

Walking through the densely packed section of the grid, where human audience members were also placed with greater density, provided a physical challenge, for the performer that formed the movement composition in this section of the piece (shown in Figure 2). This composition reflects the idea that the discretization of our environment model informs possible machine behavior, in this case, the movement primitives of stepping between each grid cell. Unwrapping each phone added complexity to the workspace, providing sophisticated computers connected to network servers, at the performer's disposal and extending the workspace beyond the visible room. Moreover, the labeled numbers on each phone provided participating audience members the chance to locate where their own phone was placed in the space, giving each audience member a second point of spatial awareness—where their own personal data were being held.

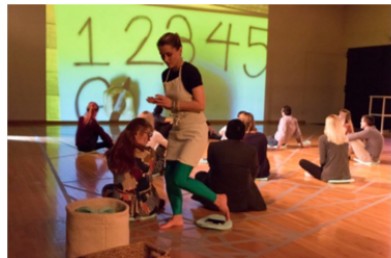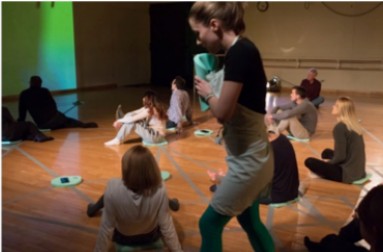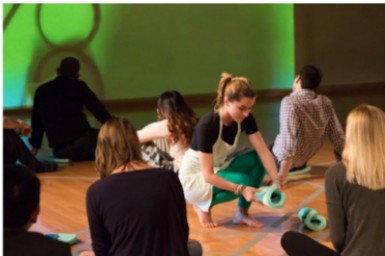

**Figure 2.** The performer ceremoniously laying out each audience member's cell phone into the segmented space during reading of Turing's excerpts. Photos by Natalie Fiol.

*4.4. Part 2: Edge Tracing and Cell Phone Sounds Solo*

Once all phones were out in the space, the video cut to a blank white projection, back-lighting the performer and the audience members within the space. For a moment, the performer stood in silence in one of the larger open grid cells. Slowly, the performer began shifting weight and tracing the edges of her body with other edges of her body. This movement composition began by probing the question "How many shapes can I make in this container?" It continued by expanding the exploration to the edges of the grid cell, asking "How many things can I do in this finite section of space?" Snapshots from this are shown in Figure 3.

During this movement composition, cell phones began ringing, an action that involved sending a command to nearby cell phone towers and distant internet servers. First, three phones (numbered 1–3) encircling the performer, which were all in close sight to their owners, began ringing. The ringtones each phone had was not known prior to the show. A Python script was used to call the phones in sequence, leveraging the web service Twillo and the `TwiML` package. More phones began ringing until, eventually, all phones were called, creating an electronic cannon for the soundscape.

Anecdotally, none of the audience members on [date redacted] answered their phones. This indicates that the impoverished setting, where participating audience members were sitting in a sparse environment with choreographed distances between them, did not encourage normal,

full-bodied reactions. Certainly, in a traditional audience setting, audience members respond richly to their phones ringing, and, in this performance, which was advertised to be interactive, these cues could have signaled action for the audience members since their phones were on visibly labeled cushions (and were probably also visibly familiar to each participant based on identifying features such as color). Similar to a Turing Machine, which does not describe the architecture of modern, practical computers, the performance created a lifted, abstract environment where audience members did not react as they typically would to familiar stimuli.

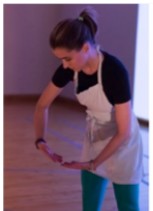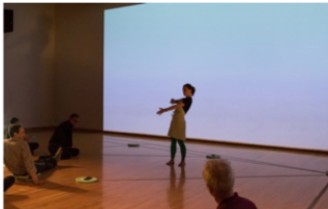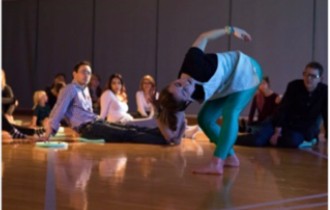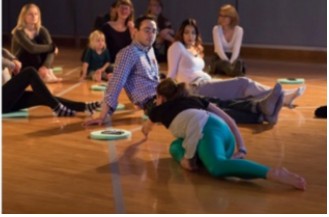

**Figure 3.** A movement section exploring the limits of the physical body and the segmented space is accompanied by the ringing of cell phones in a canon (one phone breaks silence and then is slowly joined by all 13 phones onstage). The performer stays inside one cell (this is a discrete statement), tracing its edges and finding unusual poses created by trying to move each joint in succession, trying out the Cartesian product of all joint angle ranges. Photos by Natalie Fiol.

### 4.5. Part 3: Questioning Propositional Logic Statements with "True" or "False"

The next section of the piece was demarked with distinct slide changes cued offstage such that the performer could interact with the audience members sitting in the performance space. On each slide was a phrase or symbol. A partial list of phrases is provided below and in the images in Figure 4.

- I, the performer, am a human.
- I have three arms.
- I believe I have three arms.
- We are all, at least a little, happy.
- It is not raining and I do not believe that it is not raining (Elgin 2010).
- You are answering of your own free will.
- *a black parallelogram*
- You will have a joyful reunion with your phone at the end of the show.
- I don't know the answer.

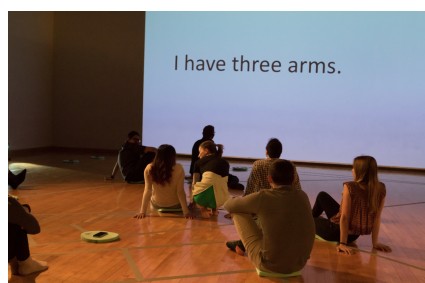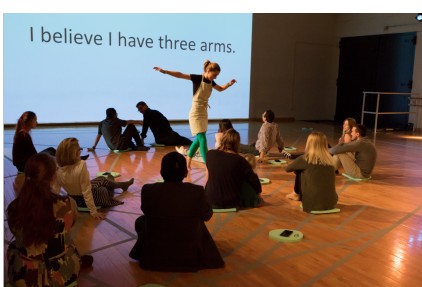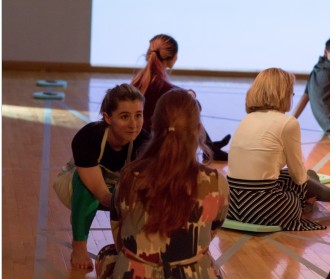

**Figure 4.** The performer questioning audience members with propositional statements. Each query takes on a unique tone—and often elicits answers loaded with quality that betray more than the requested "True" or "False" response. For example, "You will have a joyful reunion with your phone at the end of the show" may cause a dry, wry "False" or a sugary, sly "True", betraying the simultaneous desire to be returned to expensive personal property and the antagonistic relationship many have with their distracting portals to the Internet. Photos by Natalie Fiol.

Each of these phrases was displayed on the projected area and then performer would ask audience members "True or False?", effectuated with variable affect—as an urgent question or a playful joke or a cautious query. For slides that had a shape and no words, the performed heightened the approaching motion, catching the eye of the audience member and then creating a pronounced movement phrase before asking the question aloud. Thus, these propositions alternated between clear and concrete versus ambiguous and abstract.

This activity plays with the idea of whether all of human experience and behavior can be formulated in first-order propositional logic statements. It was also, by offloading the phrasing of the questions—some of which were completely non-verbal—to the body of the performer, an exercise in creating questions that don't quite have a clear cut answer as well as pointed statement to the audience about what binary choice, which drives the core of our digital machines today, does not capture. Another question was, "would people stick to this performative structure?" For example, an audience member could refuse to answer—or reply with a response outside of the two options given.

Anecdotally, in this performance the only audience member not to answer either "true" or "false" was a young toddler. Moreover, it was clear that every answer contained more information than the one bit that might be implied by the structure of the exercise. Some people answered only after a long pause or with a twinkle in their eye or an uptick at the end of their response that made their answer seem more like another question than a reply. These quirks communicated information, breaking the forced binary structure into something richer.

### 4.6. Part 4: What Does Dance Have to Offer?

The final section of the performance featured another projected video with excerpts of an academic paper read aloud. In this section, the performer read excerpts from Elgin's (2010) paper discussing exemplification in dance and began destroying the clearly written set of numbers from the previous section, creating abstract lines with varied movement qualities and rendering some symbols unreadable. The reading of the excerpt began "What, then, is dance up to? My thesis is that dance conveys understanding." and continued with Elgin's discussion of propositional ideas and the utility of dance.

Once this video had been playing for a few seconds, the performer re-entered and began walking laps of the performance space (Figure 5). This echoed the linear back and forth trips of the first section with a contrasting circularity in pathway. The performer continued stepping only in between the segmented cells, creating tiny quick steps in the densely gridded section and longer more assured strides in the less dense portion of the space.

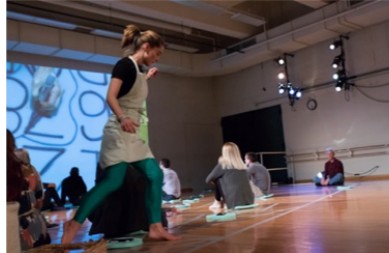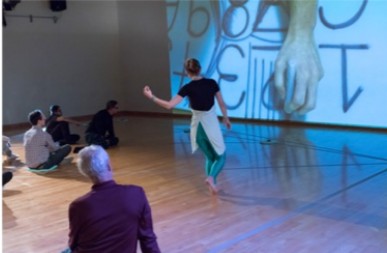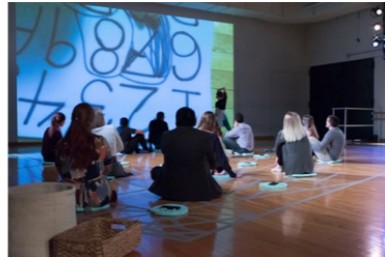

**Figure 5.** The closing sequence with excerpts of Elgin's writing read alongside a final, more integrated movement sequence. Still stepping to avoid the tape on the ground, the performer increases the speed and connectedness of pathways through space; rather than treating the tape like a constraint, she treats it like a rhythm that pushes the pace of the movement and creates a comfortable stride timing. Photos by Natalie Fiol.

After a few laps the performer began to exhibit a hand gesture with pursed fingers poised behind her head, which caused a focus change. This gesture built some urgency, suggesting the performer was being chased by a lingering thought or bad memory. Eventually the performer ended back in the largest cell of the grid where this hand gesture became circular, encircling the performer's head



accompanied by large spinal flexion and extension in a similarly circular manner. After this built, the performer exited the space and the video finished playing, ending with Elgin's description of how dance helps us understand the world through exemplification.

*4.7. Aftershow: A Descriptive, Guided Talk Back*

Finally, audience members were invited to stay for a discussion. Those who stayed formed a circle for feedback and were prompted to share what they remembered and what they experienced—rather than directive points or valued judgments of the piece. This format, derived from The Fieldwork model and shown in Figure 6, allows for a more interactive discussion of the ideas explored in the piece. Audience members especially remembered the "true/false" section of the piece and a spirited conversation about binary systems, logic, and capacity ensued from this point. Audience members also felt there must be rules governing the experience that they needed to figure out, echoing the experience of computer programming. Some related feelings of anxiety and misunderstanding between themselves and their machines (specifically their phones). Finally, several were curious to learn more about the philosophical constructs of Turing and Elgin regarding the capacity and purpose of movement.

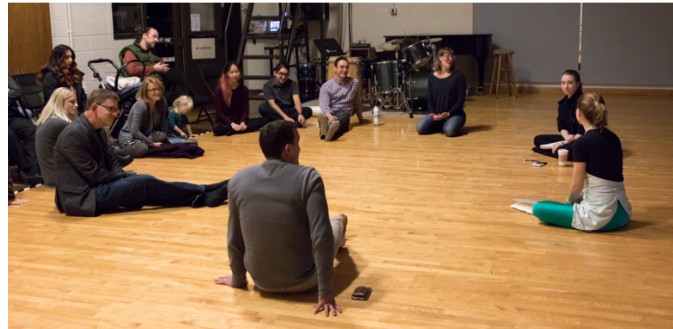 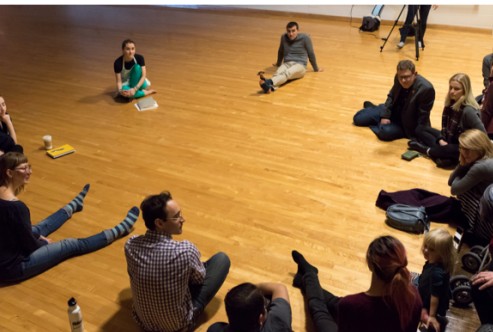

**Figure 6.** Audience members gathered for a post-show discussion. Photos by Natalie Fiol.

**5. Conclusions**

The paper has shown a relationship between mechanization and computation, pointing to an abstract model of machine behavior and extending its discussion to machines that interact with their environments. This idealized notion of a robot was then compared to the subjective experience of humans in an artistic medium. The paper has covered how these ideas were translated to a live dance performance. The goal of this work was to probe questions about the differences between humans and machines, contrasting an increasingly common practice of using machine description to describe human function.

Formalisms from computer science apply to robotics not only through explication of how the computers that control robots work; indeed, robots themselves are machines the same as computers. Thus, the ideal model for what a robot is (a machine of finite means that can interact with a segmented environment) presented here translates this idea to the articulated machines under increasing development. It also underlines the question that, if machines have behaviors, movements inside their finite form, which they cannot exhibit, do humans? Extensions of this work may consider how more advanced models of computation apply to human subjective experience. Moreover, many more artistic interpretations of these ideas may be explored, both in iterations and re-stagings of *A Machine* and in other artists' work.

Technical models derived from engineering practice may always be useful guides to dissecting the mysteries of biology. However, it is a plain, experienced and observed fact that we seem to function, qualitatively, very differently form machines. Where machines excel at repeatable execution of tasks, humans excel at variable execution of tasks; where machines excel at rapid, rote computation, humans excel at quick deduction in previously unseen scenarios; and where machines can outperform

in tasks of force/torque magnitude, humans outperform machines in communicating complex ideas through motion—as in dance. As we are not yet close to a complete, formal model of human motion, metaphors such as Turing machines—and the variation introduced here—are the only way we can grapple with representations of what humans might be. Thus, exploration through artistic and experiential methods may be important fodder in the process of understanding the differences between man and machine with greater clarity. Such methods are important aspects in the discussion around the future of work, human performance, and machine development in the 21st century.

**Author Contributions:** A.L. prepared, researched, and wrote this paper. A.L. choreographed and performed *A Machine*.

**Funding:** This work was partially funded by DARPA award #D16AP00001.

**Acknowledgments:** The author would like to thank the Dance Department at the University of Illinois at Urbana-Champaign for providing space, Erin Berl and John Toenjes for providing technical support, Reika McNish for videography, and for Natalie Fiol for photography at the showing of *A Machine*.

**Conflicts of Interest:** The author declares ownership in AE Machines, Inc. and CAAlI, LLC.

## Appendix A. Program Note

A Machine. Presented by the Robotics, Automation, and Dance (RAD) Lab.

It is difficult to distinguish life, and non-life, on purely mechanistic criteria.

—J. Scott Turner, SUNY-ESF, via email

A Machine asks audiences to push back against the machine-based metaphors we assign to humans. Are we "programmed" to behave the way we do? Is the "mechanism" of our spine malfunctioning when a slipped disc causes lower back pain? Are we replacing ourselves with robot "arms"? Do computers "learn" from data? These metaphors which confuse the action of natural, living systems with artificial, manmade tools fill our vernacular, blurring the line between human and machine. Thus, you have deposited your phone, a machine, to enter this performance. Did this perturb you? Are you anxious? In what ways is your phone superior to you? How can you outperform a robot? Can you live without your tools? Are you a machine?

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
