# Peer review of "Ideal Mechanization: Exploring the Machine Metaphor through Theory and Performance"

_arts, 2019_

Reviewer 1 Report

There are two almost entirely independent contributions in this paper. First, there is the philosophical/mathematical argument, stemming from the work of Turing. Second there is the artistic experience which explores some ideas near the philosophical argument. Ultimately, I feel both components are flawed for entirely separate reasons.

The argument as I follow it is this: Turing proved that there are limits to what we can compute, and the proof involves enumerating the computable numbers. Therefore, if we can construct machine movement in a similarly enumerated way, we can then make claims about the limits of theoretically possible machine movement. Furthermore, if machine movements have limits, it raises questions about the value of using machines as a model for humans and vice versa. 

My main concern with this argument is that there are two key qualities to the Turing machine that are not addressed in the mechanization argument: discrete numbers and infinite tape. The ae-machine presumes a finite set of motion primitives in a "discretized" space. However, the authors do not make an argument that such a thing exists. The real world is continuous. You cannot move a muscle by flexing or extending it by "one unit". The motion is continuous, starting at one point in time and one position, and ending at another. Furthermore, the world cannot be discretized merely by putting tape on the floor and calling them discrete positions. As dancers can surely recognize, even with set start and end points and times, the process from moving from one to another is open to infinite variety. One could move directly with sharp accelerations, or in an indirect path with gentle transitions.

The problem with moving Turing's argument into this continuous space (without arguing strongly that it is in fact discrete) is that the argument relies on the end "results" (whatever that may be) of the machine being different in precise ways from other movements produced by the machine. If we consider irrational numbers to represent movements that a machine cannot produce, then there is a computable number/movement that is arbitrarily close to the irrational one. In a Turing machine, these differences are discrete, and thus can have large consequences. However, in the continuous space, you would need to make the argument that the continuity is going to sufficiently change the "results".

Secondly, the infinite tape presents a problem. In the big picture, everything does halt (https://www.xkcd.com/1266/). That is to say, everything finite is computable (in some amount of time). Section 3 addresses the "finite time window" in which the machines exist. In order for the argument about the ae-machine to be convincing, you need to be able to articulate what an infinite sequence of movements practically means. If the original argument is that humans can perform things that machines cannot, but the human has a finite-time performance, the robot can produce that finite time performance because that finite number is computable. There is a space that could be explored combining this with the continuous space described above could result in something unproducable, but that is not the argument that is written in the paper. 

Moving on to the performance based contribution in the paper, I understand from experience that writing about art in an academic setting can be challenging. However, even within the wide net afforded by the idea of "exploring" the ideas of the first half of the paper, the connections between the artistic choices made and the mechanistic movement argument presented are tenuous at best. Art does not need a hypothesis and clear measurements to prove a point, but there is a difference between the performance, and writing about the performance in a paper next to a technical argument about computability. Given the context of the previous argument, I had expected a mechanical entity to try to replicate a human movement sequence. Instead, the performance seems to embody the opposite, with a human trying to replicate an impression of what a mechanical entity does. Beyond that, there seem to be many additional elements to the performance that are not explicitly tied back to the philosophical argument, such as the handling of the cell phones, the hexadecimal pillows, questionable boolean statements and many other elements. In the context of this paper, putting the performance alongside the philosophical argument, it is not sufficient to merely mirror elements of the argument in the performance. They need to be explicitly explained and justified as being included for a specific purpose to further a thesis about the possibilities of movement. Otherwise, it is an expressionistic mashup of elements from the paper that may have merit on its own, but does not merit being included in the paper. 

In terms of the actual construction of the paper, I found the authors too frequently made broad assertions without backing them up adequately. The introduction makes claims about DeepBlue and AlphaGo with no citation. "We no longer think X" is a specific claim about a general change in thinking that needs to be backed up. The second paragraph brings a claim that robots are superior, but only cites technical specifications for specific robots, which is a biased source for claims of mechanical superiority. There is also the overly broad claim that robots cannot "recreate the social behaviors of humans" which while true in many contexts, there are contexts in which robots can replicate human behavior by some metrics. There is also an entire subfield of robotics invested in biomimetics, so the two papers cited for imitating biological organisms seem plucked at random.

The shift from page 1 to page 2 is rather jarring. In one paragraph, we're talking about mechanical and biological structures as models for each other. Then suddenly, its a specific researcher's beliefs about art. Similarly, the argument about C. Elegans is out of place in Section 3. It belongs in the introduction with the other discussions of the limits of computational modeling of biology. 

Other stray notes:

I am very familiar with Turing's arguments about computation, but have not once before this paper heard them referred to as "a machines". 

I don't understand why there are only 13 cushions but you say its base 16. 

The text of the paper refers to the Figures multiple times, but each contains 10 photos. These should be broken up into subfigures so that individual pictures can be referred to, i.e. "...corresponding to a numbered cushion (shown in Figure1c)"

Reviewer 3 Report

The paper presents a dance performance that is conceptually based on considerations about the human-machine metaphor, and more precisely on an adaptation of Turing’s concept of an a-machine (Turing machine).

In the introduction, a clear line of thought is missing in my view, and questions at the end of each paragraph are not always well motivated. Biomimetic models should play a more prominent role here in this line of argument, but it should be clear what they model, what their purpose is (are they purely explanatory? Or do they serve other functions as well?), and what they are not supposed to be (see. e.g., Braitenberg’s Vehicles; von Neumann, Rolf Pfeifer, ...). More relevant literature from this field should be applied to make the argumentation more convincing.

I do not see the necessity of assuming that, as next step, biological systems should imitate machines... what could be interesting is the question if explanatory models „override“ their purpose and become primary tot he systems they are supposed to explain in the minds of „users“, or the public (e.g., many computer scientists and students are not aware that artificial neurons are very reduced models of biological neurons, and their parts and features have biological equivalents that are by far more complex).

Conclusions: what has been gained from the performance? Which insights have been produced by staging it, apart from those that have motivated the performance from the start? What are outcomes and potential next steps?

Line 3: „exploration exploring“  - please rephrase.

Lines 5-8: this sentence is not clear – please check grammar, and rephrase.

l. 7: what is meant by „their machines“

l. 8: „the result ... is“ or „the results ... are“

l. 18 as => has

ll. 17-20: is there a reference for this statement? Who say that we (who is „we“?) think this way?

ll. 20-21: this question does not necessarily follow from the previous – why is this question relevant? What bodily functions are referred to?

l. 29 this question should be better motivated.

ll. 34-35: optimality has not been proposed as model, but rather as criterion for (e.g.) modelling motor control (not really clear from this quote what is meant by optimality...)

ll. 30-36: please be more precise here – what is modelling (mimicking, ...) what?

ll. 51-53: how does product design come in here? I do not see how this may contribute to the line of argument.

ll. 58-60: what is said here about dancers is rather general, and does not require the relation to robotics or artificial systems. Please make the line of argument more clear and comprehensible!  What is the message of this paragraph? Why is the question for limits relevant in this context?

l. 59: more versatile instruments – of what?

L. 61 „of robotic systems“ is doubled – please delete one.

l. 66 delete „the“

ll. 66-67: questions could be phrased more consistently with the line of argument – why are these questions interesting? How are they motivated?

l. 68: please clarify: what is meant by „ an abstract, discrete ideal of mechanical motion“?

ll. 69-71: From this short outlook, it is not clear how that different parts oft he paper come together. What ist he over-all motivation of questioning the explanatory power of technical (biomimetic) models and solving this problem by means of dance performance?

ll. 73-75: not sure that this has been done... please be more precise. Why would you say that movement is inherently part of computation (in general)?

l. 85: „set up for desired behaviour“ – not clear.

l. 91: which / whose future work?

l. 93: capacity of machines – to do what?

l. 113: how does the concept of motion primitives come in at this point? Please give references, or clarify what this is based on

l. 132: „that fire together“ - are motion primitives conceptualized as neurons here? Please clarify.

ll. 142-143: say more about Moore’s Law, this is not self-explanatory.

l. 144: which benchmarks?

ll. 144-145: is this the case? Why? And how?

l. 146: the same set as?

l. 147: „the functioning of brains and motor control systems is not entirely known“ – this statement is so broad, general and vague that it fails to make any meaningful point. A lot is known indeed about brains and motor control systems, and much more than is relevant in the context oft he article, so please be more precise, and if possible give references.

ll. 149-154: correlation between what exactly? Please rephrase for clarity (and correct grammar).

ll. 163-165: these questions are very diverse, and not clearly motivated...

l. 172: what is „philosophy of mind versus machine“? not clear, please rephrase.

ll. 183-186: not clear, please rephrase.

l. 211: how were the participating audience members pre-elected? Please explain the process, criteria, conditions, etc.

l. 224 were?

l. 230 „several moments“ – please rephrase

l. 253 a circle of three phones?

l. 294: why is this a dialogue?

l. 296: „a few moments2 – please rephrase

ll. 321-323: I doubt that the paper has done that yet – it could do, if the Introduction has been improved

Reviewer 4 Report

The paper is definetly an interesting paper.

The confrontation between Turing-like theoretical models of computation and life-based artistic performances is enlightening. Experiencing such a confrontation is a way for people to better understand the real status of today technology. Both parts of the paper are clear and well balanced.

Two minor remarks:

1- The focus on dance would deserve to be better developed. The following monograph contains many chapters that can be interesting for you to introduce in your bibliography: "Dance Notation and Robot Motion", STAR Series 111, Springer, 2016.

2- Your conclusion is fair and honest. Why not to push a little bit more towards the underlying idea that humans definetly do not behave like machines? And then, go back to the status of  metaphors (announced in the title) which are the only way to speak about machine performance: metaphors allow to provide mental representations which are not "mechanizable". In that sense they are more powerfull than algorithms.

Commenting these two issues would improve the paper.

Author Response

Thank you so much for these insightful comments. Point 1 has been address in Lines 109-113; Point 2 (how poetic!) has been addressed in Lines 481-484. These changes have meaningfully improved the manuscript.

Round  2

Reviewer 2 Report

The author(s) have made a number of modifications which have improved the paper, however at the heart of the problematic that forms this second review is a blurring from the author(s) as to what the paper is intended to address:

On the one hand, in its opening paragraph, the author(s) states: "The paper uses known limits on computation, previously proved by Turing, to model the process of mechanization, machines interacting with an environment" whereas the submitted response to reviewers the author(s) explicitly clarifies, "I apologize for the poor writing which led the reader to believe that the paper discusses real computers or real robots". In other words, the the æ-machines is designed to serve as an abstraction, "an idealization" of real robots and real robotic movement .. 

.. and certainly, in the opinion of this reviewer, this concept - the ae-machine as a model of embodied [robotic] movement - is a beautiful, elegant and insightful device that is worthy of investigation, theoretization and dissemination; particularly in the light of real dancer's feedback as they dance with real robots, "... a lack of variability in the motion of these systems; correspondingly, choreographers have found the platforms to be frustratingly limited in their expressive capacity. The functionally efficient approach used by roboticists to design robot movement may be part of this experience. In contrast, dancers care about expressively rich movement, working to create many options for movement behaviors in their bodies through extensive training".

Nonetheless, in the author(s) response it is clarified that the performance is not intended as a literal representation of an ae-machine, but rather the performance is used to "probe ideas [of the machine] in an abstract way", indeed, the goal of the setting to create a sense that "one is inside a machine". I.e. The goal of the performance is to bring forth the experience that we are inside of [contra looking at] the ae-machine.

This is clearly a worthy goal, particularly in the context of questioning the mechanical metaphor in dance, however in any theatrical protocol that derives from [even the abstract] deployment of the ae-machine metaphor, it is [ex hypoothesi] essential - if the metaphor is to have traction at all - that there is no input to the system that holds any possibility of subsequently modifying the behaviour of any of its component elements (beyond that inertly defined and contained in the machine's initial workspace). If this is not the case - e.g. the experimental work described features ringing on participant's phones [which may cause them to get up and answer them, albeit, perhaps tellingly, no one in the performance did] - then the author(s) are more formally obligated to weave their paper around Wegners' "Interaction Machine" metaphor (as discussed in the first review).

In summary, although the paper is certainly improved after its thorough revision, it continues to be formed by two, almost independent, sections: (i) the ae-machine as an abstraction for embodied mechanical movement [qua non-interaction] and (ii) the theatrical explorations that this conception raises as to how - if at all - real, embodied human movement steps beyond the underlying Turing model. Unfortunately, in the presented work it is remains unclear how the experimental protocols described explore (ii) nor - in the light of 'non-closed-system' nature of the described performance - precisely how the ae-machine metaphor is intended to hold. 

Nonetheless, these questions are foundationally interesting; indeed, I believe the two elements would work much better as two distinct contributions: the first, theoretical, exploring the ae-machine as a model of embodied movement, perhaps with a case study detailing how this might be realised in, say, defining a simple artistic movement in performance and the second theatrically exploring, from an experiential [first person?] perspective, a simple Finite State Automaton as it controls either Turing's a-Machine [or the author's ae-machine] as it undertakes a simple computation (or simple movement). Of course, in both cases some theatrical preamble would be required to illustrate the operation of an FSA (cf. Turing's simple DSM mechanism as detailed in the 1950 Mind paper) but in both, issues around the environment, non-participatory interactions, boolean responses etc. could continue to be explored along the lines the author(s) have imagined.

Author Response

Dear Reviewer,

Thank you very much for this thoughtful review.  It is wonderful to read about how the paper has inspired the reviewer to consider many additional facets of computer science, robotics, and the arts.  This is the purpose of the paper!  From reading this reviewer's reviews, it is clear that the reviewer has many wonderful ideas about future work that could, indeed, build upon the work discussed in the manuscript.  To further clarify the points regarding Wegner's work and to improve the flow of the manuscript, Lines 168-175, have been edited.  I hope this addresses the reviewers' confusion.  Again, it is the enumeration of computable sequences that render Wegner and Turing's metaphors equivalent from the narrow point of view of this work (see added citation [52]).

Sincerely,

The lead author